# Plasma Trimethylamine-N-oxide and impaired glucose regulation: Results from The Oral Infections, Glucose Intolerance and Insulin Resistance Study (ORIGINS)

**Sumith Roy**[1], **Melana Yuzefpolskaya**[2], **Renu Nandakumar**[3], **Paolo C. Colombo**[2], **Ryan T. Demmer**[1,4]*

1 Department of Epidemiology, Mailman School of Public Health, Columbia University, New York, NY, United States of America, 2 Division of Cardiology, Department of Medicine, New York Presbyterian Hospital, Columbia University, New York, NY, United States of America, 3 Irving Institute for Clinical and Translational Research, Columbia University Medical Center, New York, NY, United States of America, 4 Division of Epidemiology and Community Health, School of Public Health, University of Minnesota, Minneapolis, MN, United States of America

* demm0009@umn.edu

**Data Availability Statement:** All relevant data are within the paper and a limited data set is available on figshare at doi: 10.6084/m9.figshare.9913667.

## Abstract

Trimethylamine-N-oxide (TMAO)–a gut-microbiota metabolite–is a biomarker of cardiometabolic risk. No studies have investigated TMAO as an early biomarker of longitudinal glucose increase or prevalent impaired glucose regulation. In a longitudinal cohort study, 300 diabetes-free men and women (77%) aged 20–55 years (mean = 34±10) were enrolled at baseline and re-examined at 2-years to investigate the association between TMAO and biomarkers of diabetes risk. Plasma TMAO was measured using Ultra Performance Liquid Chromatography-Mass Spectrometry. After an overnight fast, FPG was measured longitudinally, HbA1C and insulin were measured only at baseline. Insulin resistance was defined using HOMA-IR. Multivariable generalized linear models regressed; i) FPG change (year 2 minus baseline) on baseline TMAO tertiles; and ii) HOMA-IR and HbA1c on TMAO tertiles. Multivariable relative risk regressions modeled prevalent prediabetes across TMAO tertiles. Mean values of 2-year longitudinal FPG±SE across tertiles of TMAO were 86.6±0.9, 86.7 ±0.9, 86.4±0.9 (p = 0.98). Trends were null for FPG, HbA1c, HOMA-IR, cross-sectionally. The prevalence ratio of prediabetes among participants in 2nd and 3rd TMAO tertiles (vs. the 1st) were 1.94 [95%CI 1.09–3.48] and 1.41 [95%CI: 0.76–2.61]. TMAO levels are associated with increased prevalence of prediabetes in a nonlinear fashion but not with insulin resistance or longitudinal FPG change.

## Introduction

Type 2 diabetes is an important public health problem with over 400 million diagnosed cases globally, and in the United States the prevalence of diagnosed diabetes increased from 0.93%

**Funding:** This research was supported by NIH grants R00 DE018739, R21 DE022422 and R01 DK 102932 to Dr. Demmer. This publication was also supported by the National Center for Advancing Translational Sciences, National Institutes of Health, through Grant Number UL1TR001873. The content is solely the responsibility of the authors and does not necessarily represent the official views of the NIH. The funders had no role in study design, data collection and analysis, decision to publish, or preparation of the manuscript.

**Competing interests:** The authors have declared that no competing interests exist.

in 1958 to 7.40% in 2015[1]. Similarly, impaired glucose regulation (i.e., prediabetes) is also a growing public health concern. In 2012, an estimated 86 million people in the U.S. aged 20 and older had prediabetes[1], which is a strong preclinical risk factor for future type 2 diabetes. A better understanding of disease susceptibility and environmental risk factors is needed to address the growing burden of type 2 diabetes.

The microbes inhabiting the gastrointestinal tract have been hypothesized to play an etiologic role in the development of cardiometabolic diseases. Recently, Le Chatelier et al.[2] found the gut microbiome to be associated with adverse metabolic profiles both cross-sectionally and longitudinally among diabetes-free individuals, bolstering the potential for the gut microbiota to contribute to early diabetes risk, although the mechanisms remain uncertain.

Trimethylamine-N-oxide (TMAO)–a gut microbiota derived metabolite–has been hypothesized as risk factor for cardiometabolic disease. TMAO is produced primarily by the metabolism of dietary nutrients such as choline, phosphatidylcholine and L-carnitine by intestinal bacteria to produce trimethylamine which is subsequently converted to TMAO in the liver. There is strong evidence linking circulating TMAO levels to increased risk for myocardial infarction and stroke[3, 4], even in low risk individuals (e.g., age <65 years, women, low lipid levels, low C-reactive protein (CRP) levels)[5]. In regard to diabetogenesis, animal models have shown that dietary TMAO can lead to impaired glucose tolerance, increase fasting insulin levels and adipose tissue inflammation, in mice fed a high fat diet[6]. Others have demonstrated that knockdown of flavin containing monooxygenase 3 (FMO3)–which produces TMAO–in insulin resistant mice blocks the development of hyperglycemia[7]. In humans, a few studies have also shown increased TMAO levels to be associated with type 2 diabetes. Most prior studies[8–11] were cross-sectional precluding the ability to determine whether elevated TMAO preceded type 2 diabetes development or resulted from the diabetes phenotype which is often associated with alterations in host physiology, including nephropathy, which influences TMAO levels via reduced renal clearance. Additionally, health behavior changes and new pharmacological therapies occurring in response to type 2 diabetes diagnosis could contribute to intestinal dysbiosis and increased capacity for TMAO production. To our knowledge, only one longitudinal study has explored baseline TMAO as a predictor of future diabetes development [12] and, surprisingly, they report elevated TMAO levels to be related to decreased type 2 diabetes risk. Therefore, limited data are available exploring the value of TMAO in predicting type 2 diabetes development. No existing studies have explored the relationship between TMAO and early risk biomarkers linked to future type 2 diabetes development such as insulin resistance, rising longitudinal glucose levels or prediabetes.

Presently, we have studied the relationship between plasma TMAO and early biomarkers of type 2 diabetes risk. We hypothesize that elevated plasma TMAO-levels would be associated with markers of insulin resistance and impaired glucose regulation, cross-sectionally, as well as with rising glucose levels, longitudinally. These investigations are undertaken in a diabetes-free population without a history of cardiovascular and/or kidney disease.

## Materials and methods

### Study population

The Oral Infections, Glucose Intolerance and Insulin Resistance Study (ORIGINS) is a longitudinal cohort study investigating the relationship between subgingival microbial community composition, systemic inflammatory phenotype and impaired glucose metabolism[13]. The current analysis includes the first 300 participants enrolled from February 2011 to May 2013. Participants were recruited via postal mailings, email blasts, posted flyers, information sessions and word-of-mouth strategies.

Inclusion criteria were as follows: Men and women aged 20–55 years without: i) Diabetes Mellitus based on self-report physician diagnosis, fasting plasma glucose (FPG) $\geq$126 mg/dl or hemoglobin A1c (HbA1c)$\geq$6.5% (48 mmol/mol); ii) self-reported history of myocardial infarction, congestive heart failure, stroke or chronic inflammatory conditions. Participants were examined at baseline and n = 297 had a TMAO assessment, and n = 241 (81%) provided fasting blood at a two-year follow-up visit (February 2013 to December 2015[13]). The Institutional Review Boards of Columbia University and The University of Minnesota approved the study. All participants provided written informed consent.

## Trimethylamine-N-oxide assessment

The exposure Trimethylamine-N-oxide (TMAO) was only measured once at baseline in human plasma samples using Ultra Performance Liquid Chromatography-Mass Spectrometry (UPLC-MS/MS) after protein precipitation using deuterated (D9)-TMAO as the internal standard[14]. UPLC-MS/MS analysis was performed on a platform comprising Eksigent ULC 100 integrated to API 4000 mass spectrometer controlled by Analyst 1.6 (ABSciex, Foster City, CA).

## Laboratory measures

At baseline, FPG, serum insulin and lipids, and HbA1c were measured from blood after an overnight fast using a Cobras Integra 400 Plus (Roche Diagnostics, Indianapolis, IN, USA) as previously described[15, 16]. The Homeostatic Model Assessment of Insulin Resistance (HOMA-IR) was used to define insulin resistance as previously defined[17, 18]. Baseline prediabetes status was defined based on either one of the below criteria being fulfilled: HbA1C of 5.7–6.4% (39–46 mmol/mol) or FPG between 100 and 125 mg/dl[19]. FPG was also measured at the year 2 follow-up visit.

## Risk factors

Cardiometabolic risk factors were measured by trained research assistants in space provided by a Center for Translational Science Award (CTSA). Seated systolic and diastolic blood pressures were measured in triplicate and the last two measurements were averaged. Participant body mass index (BMI) was calculated as weight in kilograms/height in meters$^2$. Questionnaires were administered to obtain information on: age, sex, race/ethnicity (non-Hispanic Black, non-Hispanic White, Hispanic, Other), educational level (high school completion, college or vocational training, advanced degrees), cigarette smoking (current, former or never smoking and duration/intensity of smoking). Leisure-time physical activity (LTPA) was assessed and activities were converted into metabolic equivalents (METS), further categorizing them into four LTPA categories in accordance with the 2008 Physical Activity Guidelines for Americans: no LTPA reported, low (0 to <500 MET min/wk), moderate(500 to <1,000 MET min/wk), high($\geq$1000 MET min/wk)[20, 21].A detailed food frequency questionnaire was administered from which The Alternative Healthy Eating Index (AHEI) score was calculated to represent diet quality based on the intake of 9 components: vegetables, fruit, nuts and soy, white or red meat, *trans*fat, polyunsaturated or saturated fat, fiber, multivitamin use, and alcohol[22]. A higher total score of AHEI indicates a lower risk of developing chronic disease particularly chronic heart disease and diabetes[22, 23].

## Statistical analysis

All statistical analyses were conducted with SAS 9.4. (SAS Institute, Cary, NC). Difference in means or prevalence of potential confounders according to TMAO levels or metabolic

variables (i.e., insulin resistance, FPG and HOMA-IR) were assessed using one-way ANOVA for continuous variables and chi-square for categorical variables. Multivariable linear models regressed natural log transformed insulin resistance (due to non-normality of the original variable), FPG or HbA1c on tertiles of TMAO, in separate regression models. TMAO was divided into tertiles to relax linearity assumptions. Sequentially adjusted regression models were formed to assess the degree of confounding by specific sets of confounders. Model 1 was unadjusted. Model 2 was adjusted for age, gender, race/ethnicity, and education. Model 3 was further adjusted for BMI, systolic blood pressure and HDL. Model 4 was further adjusted for

AHEI. Tests for linear trends were performed using TMAO as a continuous variable in the aforementioned regression models. A multivariable modified Poisson regression with robust error variance was used regressed prediabetes prevalence across tertiles of TMAO; prevalence is defined as the probability of having prediabetes. Prevalence ratios and 95% confidence intervals (95%CI) are presented for the 2nd and 3rd tertiles of TMAO levels relative to the 1st tertile.

## Results

### Baseline characteristics

Participants had a mean age of 34±10 years, and 77% were female. Median plasma TMAO level and Interquartile range (IQR) are 2.69 μM and 1.9–4.22 μM, respectively. TMAO levels were modestly associated with increased age, Hispanic ethnicity and BMI (Table 1). Interestingly, diet quality as assessed by AHEI did not differ by TMAO level. Similarly, the AHEI subscore corresponding to meat consumption was not related to TMAO levels. Additional participant characteristics are summarized in Table 1.

### Cross-sectional associations between TMAO and biomarkers of diabetes risk

TMAO levels did not explain variation in FPG, HbA1c or HOMA-IR cross-sectionally, as summarized in Table 2. Results were very consistent across varying degrees of multivariable adjustment (Table 2). After full multivariable adjustment (model 4, S1 Fig), the prevalence ratio of prediabetes among participants in the 2nd and 3rd TMAO tertiles (vs. the 1st) were 1.94 [95%CI 1.09–3.48] and 1.41 [95%CI: 0.76–2.61] and results were consistent across multivariable models (S1 Fig). When combining participants in the 2nd and 3rd tertiles, the prevalence ratio for prediabetes was 1.71, p = 0.05 in crude models although results were attenuated and lost statistical significance after multivariable adjustment (S2 Fig).

### Longitudinal association between TMAO and fasting plasma glucose

There was no statistically significant association between baseline TMAO and follow-up FPG. In unadjusted models, mean follow-up FPG across tertiles of TMAO were 86.8±1, 86.9±1, 87.1 ±1 mg/dL respectively. Results were consistently null in multivariable models (S3 Fig).

## Discussion

We report that TMAO levels were not associated with insulin resistance, HbA1c or fasting plasma glucose cross-sectionally, or with longitudinal change in fasting plasma glucose. TMAO levels were associated with a modest increase in the prevalence of prediabetes in a non-linear fashion such that participants with intermediate TMAO levels had a statistically significant 94% increase in prediabetes prevalence. These results were generally consistent regardless of level of risk factor adjustment although statistical significance was lost for participants in the highest TMAO tertile.

**Table 1. Participant characteristics overall and by TMAO Tertiles: (ORIGINS) 2011–2015.**

| | All (N = 297) | Tertile 1 (n = 99) | Tertile 2 (n = 99) | Tertile 3 (n = 99) | P Value |
|---|---|---|---|---|---|
| TMAO (median, range) | 2.69(1.90–4.22) | 1.73 (1.411.90) | 2.69(2.33–2.98) | 5.52(4.22–7.66) | N/A |
| **Age,years** | 34.06±9.86* | 32.28±0.92 | 33.87±0.98 | 36.01±1.03 | 0.02 |
| Female | 77.10% | 78.79% | 77.78% | 74.75% | 0.78 |
| **Race** | | | | | 0.007 |
| Hispanic | 46.80% | 49.50% | 34.34% | 56.57% | |
| Non-Hispanic White | 22.90% | 23.23% | 26.26% | 19.19% | |
| Non-Hispanic Black | 16.84% | 11.11% | 20.20% | 19.19% | |
| Other | 13.46% | 16.16% | 19.20% | 5.05% | |
| **Education** | | | | | 0.5 |
| < college | 31.99% | 26.26% | 32.32% | 37.38% | |
| 4 years of college | 45.45% | 51.52% | 44.45% | 40.40% | |
| >college | 22.56% | 22.22% | 23.23% | 22.22% | |
| **Activity level** | | | | | 0.59 |
| None | 30.58% | 31.26% | 32.65% | 27.84% | |
| Low | 12.03% | 8.33% | 13.27% | 14.43% | |
| Moderate | 16.15% | 14.58% | 19.39% | 14.43% | |
| High | 41.24% | 45.83% | 34.69% | 43.30% | |
| AHEI Score | 49.05±11.88* | 48.3±1.2 | 50.2±1.3 | 48.7±1.3 | 0.53 |
| AHEI meat score | 6.20±3.50* | 6.6±0.4 | 6.0±0.4 | 6.0±0.4 | 0.41 |
| BMI (kg/m$^2$) | 27.07±6.13* | 26.83±0.61 | 26.08±0.52 | 28.29±0.68 | 0.03 |
| **Body Mass Index category** | | | | | 0.04 |
| Normal | 44.44% | 50.51% | 50.51% | 32.32% | |
| Overweight | 32.33% | 28.28% | 31.31% | 37.37% | |
| Obese | 23.23% | 21.21% | 18.18% | 30.30% | |
| Systolic blood pressure,mm Hg | 117.75±12.45* | 117 ±1.21 | 117±1.33 | 119±1.20 | 0.52 |
| Diastolic blood pressure,mmHg | 75.25 ±9.71* | 75 ±0.91 | 75 ±1.05 | 75 ±0.96 | 0.98 |
| Hypertension | 97 (32.66%) | 29 (29.29%) | 33(33.33%) | 35 (35.35%) | 0.65 |
| Prediabetes | 17.85% | 12.12% | 21.21% | 20.20% | 0.19 |
| FPG (mg/dl) | 85.22±7.64* | 85.23±0.82 | 84.15±0.69 | 86.28±0.77 | 0.15 |
| HbA1c (%)/ mmol/mol | 5.36±0.34* (35±3.7)* | 5.32±0.03 (35±0.3) | 5.36±0.03 (35±0.3) | 5.39±0.03 (35±0.3) | 0.38 |
| Total cholesterol (mg/dl) | 172.61±30.74* | 173.56±3.13 | 174.22±3.04 | 170.06±3.10 | 0.59 |
| LDL-cholesterol (mg/dl) ** | 97.98±27.86* | 99.98±2.96 | 98.33±2.79 | 95.63±2.66 | 0.54 |
| HDL (mg/dl) | 59.05±16.06* | 58.26±1.60 | 59.84±1.58 | 59.04±1.66 | 0.79 |
| Chol to HDL ratio | 3.12±0.05 | 3.18±0.11 | 3.10±0.09 | 3.08±0.09 | 0.75 |
| Triglyceride (mg/dl) | 77.80±45.50* | 77.12±3.80 | 79.61±5.24 | 76.66±4.59 | 0.89 |
| Insulin (median, 25th 75th percentile) | 8.8(5.9,12.0) | 8.5 (5.8,12.5) | 8.0 (5.6,11.3) | 9.7 (6.7,12.3) | 0.14 |
| HOMA-IR (median, 25th 75th percentile) | 0.57(0.19,0.97) | 0.54(0.15,1.03) | 0.47(0.13,0.82) | 0.70(0.31,1.04) | 0.08 |

*Standard deviation

**n = 4 participants missing LDL-cholesterol

To our knowledge, the current study is the first to investigate the association between plasma TMAO and early biomarkers of diabetes risk among participants free of diabetes and clinical cardiovascular disease. Our null findings might appear at odds with several studies reporting that elevated TMAO levels predict increased risk for chronic kidney disease, myocardial infarction, stroke and heart failure[5, 24]. However, it is important to note that among studies with positive TMAO findings, the enrolled participants were generally older and had

**Table 2. Mean fasting plasma glucose, HbA1c, HOMA-IR across TMAO Tertiles: Cross-sectional results from ORIGINS) 2011–2015.**

| TMAO Tertiles | FPG (mg/dl)mean±SE | HbA1c (%)/(mmol/mol) mean±SE | HOMA-IR mean±SE |
|---|---|---|---|
| Tertile 1 (n = 99) | | | |
| TMAO range (0.24–1.90) | | | |
| Model 1 | 85.23±0.76 | 5.32±0.03 (35±0.3) | 0.63±0.05 |
| Model 2 | 85.49±0.70 | 5.34±0.03 (35±0.3) | 0.64±0.05 |
| Model 3 | 85.57±0.67 | 5.36±0.02 (35±0.2) | 0.64±0.05 |
| Model 4 | 85.68±0.71 | 5.33±0.02 (35±0.2) | 0.66±0.05 |
| Tertile 2 (n = 99) | | | |
| TMAO range (1.91–2.69) | | | |
| Model 1 | 84.15±0.76 | 5.37±0.03 (35±0.3) | 0.52±0.05 |
| Model 2 | 84.20±0.70 | 5.37±0.03 (35±0.3) | 0.54±0.05 |
| Model 3 | 84.12±0.67 | 5.36±0.02 (35±0.2) | 0.56±0.05 |
| Model 4 | 84.25±0.72 | 5.34±0.02 (35±0.2) | 0.54±0.05 |
| Tertile 3 (n = 99) | | | |
| TMAO range (2.70–4.22) | | | |
| Model 1 | 86.28±0.76[a] | 5.39±0.03 (35±0.3) | 0.70±0.05[a] |
| Model 2 | 85.96±0.71 | 5.37±0.03 (35±0.3) | 0.66±0.05 |
| Model 3 | 86.00±0.68 | 5.36±0.02 (35±0.3) | 0.66±0.05 |
| Model 4 | 86.01±0.74 | 5.35±0.03 (35±0.3) | 0.66±0.05 |

Model 1 = unadjusted; Model 2 = age, gender, race/ethnicity, education

Model 3 = M2+ BMI, systolic blood pressure, HDL

Model 4 = M3+AHEI

[a]p-value for comparison of mean values between tertile 3 vs. tertile 2≤0.05

The sample size for model 4 is n = 266 for all outcomes due to missing data on AHEI.

evidence of substantial pre-existing cardiovascular disease. For example, in the elegant publications by Tang and colleagues, study participants were recruited from elective diagnostic cardiac catheterization[8] which is an indication for suspected atherosclerotic coronary artery disease and an adverse cardiovascular risk profile. Accordingly, those participants had a mean FPG in the prediabetes range (102 mg/dl), 32% had diagnosed diabetes (~double the national rate) and 72% were hypertensive[5]. Participant characteristics were consistent in a second report among a similar patient population enrolled from recipients of elective cardiac catheterizations in which TMAO was predictive of all-cause mortality among patients with chronic kidney disease[24].

In contrast, our results from ORIGINS are consistent with recent null findings among 817 participants in the Coronary Artery Risk Development in Young Adults (CARDIA study[25] which found no association between TMAO and coronary artery calcification incidence or progression. Additionally, CARDIA also reported no association between TMAO level and cross-sectional insulin resistance. As suggested by the CARDIA investigators, the younger age (~40 years) and lower cardiovascular risk (~4% prevalent diabetes and 10% using hypertensive medications) of their cohort might explain their null finding[25]. The mean age of ORIGINS participants is similarly young (34 years) while the mean age in prior positive studies was 66 years[24] and 63 years[5]. Our observation that TMAO was modestly related to increased prediabetes prevalence supports the notion that TMAO levels might only be a predictive biomarker in populations with early or established cardiovascular risk.

Renal function might also provide some level of explanation for discordant findings in our current data as compared to other cohorts. The fact that higher renal function increases

TMAO clearance raises the strong potential for confounding. Specifically, it is possible that reduced renal function causes both elevated TMAO levels and clinical cardiovascular disease (CVD) events. Our observation of increased TMAO among older participants provides indirect support for this notion. As such, our adjustment for age in multivariable models helps to mitigate confounding by renal function although future studies with assessment of renal function will be important.

It is also possible that the adverse impact of TMAO on cardiometabolic risk, only begins above a threshold of circulating TMAO that is achieved in the context of reduced renal function enabling excessive TMAO accumulation. Interestingly, Tang and colleagues found that among patients without chronic kidney disease (CKD), the predictive value of TMAO for all-cause mortality was substantially diminished and only statistically significant in the 4th TMAO quartile[24]. Moreover, the observed range of TMAO values in the non-statistically significant 1st– 3rd quartiles of TMAO were similar the ranges observed in both CARDIA and ORIGINS. Median (IQR) plasma TMAO levels in ORIGINS are 2.7 µM (1.9–4.22 µM) and in CARDIA median (IQR) values were 2.6 (1.8–4.2).[25]

Some important limitations should be noted. The ORIGINS cohort is not a nationally representative sample, limiting the generalizability of our findings. Nevertheless, the consistency in TMAO distributions between ORIGINS and CARDIA, a much larger multi-center, population-based study, suggests that results in ORIGINS are robust. Second, our sample size was small and had limited power to rule out associations of very modest magnitude. Nevertheless, the magnitude of associations observed presently are unlikely to be clinically meaningful even if larger studies identified statistically significant findings of similar magnitude. Our measure of diet quality was based on a single food frequency questionnaire which potentially mischaracterized TMAO dietary precursors proximal to the assessment of TMAO. Dietary information in the study was calculated using AHEI[22], which is based on foods and nutrients predictive of chronic disease risk generally and not TMAO specifically. Regardless, since TMAO is hypothesized as an intermediate mechanism linking diet to cardiometabolic outcomes, the dietary assessment poses minimal threat to the validity of our TMAO findings. In future studies, the precision of hypothesis tests would be enhanced by measuring gut microbiome and dietary TMAO precursors carefully, and analyzing the potential for interactions between the gut microbiome and diet on TMAO levels and subsequent cardiometabolic disease.

A strength of our study was the ability to examine the prospective association of baseline plasma TMAO levels and longitudinal changes in fasting plasma glucose among a healthy study sample which precludes potential reverse causation by the diabetes phenotype and reduces potential confounding by reduced renal function.

In a cohort of participants free of diabetes and clinical cardiovascular disease, we observed no association between TMAO levels and continuous biomarkers of early diabetes risk cross-sectionally or longitudinally. In contrast, intermediate TMAO levels were modestly associated with increased risk of prevalent prediabetes after multivariable adjustment. Future longitudinal studies are necessary to determine if TMAO increases risk for incident prediabetes and/or diabetes, despite no evident relationships with insulin resistance. Until such studies are performed, the role of TMAO as a biomarker of diabetes risk remains uncertain.

## Supporting information

**S1 Fig. Association between Tertiles of TMAO and Prediabetes Prevalence.** Model 1 = unadjusted; Model 2 = age, gender, race/ethnicity, education; Model 3 = M2+ BMI, systolic blood pressure, HDL; Model 4 = M3+alternative healthy eating index. *p<0.05.
(TIF)

**S2 Fig. Association between TMAO (Tertiles 2 and 3 vs. 1) and Prediabetes Prevalence.**
Model 1 = unadjusted; Model 2 = age, gender, race/ethnicity, education; Model 3 = M2+ BMI,
systolic blood pressure, HDL; Model 4 = M3+ alternative healthy eating index.
(TIF)

**S3 Fig. Association between Baseline TMAO Tertiles and Longitudinal Fasting Plasma
Glucose.** Model 1 = unadjusted; Model 2 = age, gender, race/ethnicity, education; Model
3 = M2+BMI, systolic blood pressure, HDL+ alternative healthy eating index; Model 4 = M3+
baseline glucose. Y axis is centered on the mean value observed in the total population
(mean = 85 mg/dL) and the range is set to twice the standard deviation of fasting glucose.
(TIF)

## Acknowledgments

We thank the following individuals for their invaluable contributions to this research: the 1199
SEIU, HS-3/SSA Area leadership including Ms. Consuelo Mclaughin, Mr. Bennett Batista, Mr.
Victor Rivera; Ms. Romanita Celenti for her efforts in performing phlebotomy and processing
and analyzing plaque samples; Ms. Aleksandra Zuk and Garazi Zulaika for their leadership
and excellent program coordination; Drs. Nidhi Arora, Ashwata Pokherel, Publio Silfa &
Thomas Spinell for their skilled examinations and essential participant engagement. We are
also profoundly grateful to the ORIGINS participants, for their participation in this research.

## Author Contributions

**Conceptualization:** Sumith Roy, Ryan T. Demmer.

**Funding acquisition:** Ryan T. Demmer.

**Writing – original draft:** Sumith Roy, Ryan T. Demmer.

**Writing – review & editing:** Sumith Roy, Melana Yuzefpolskaya, Renu Nandakumar, Paolo C.
Colombo, Ryan T. Demmer.

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
