## [Decision Letter · Decision Letter 0]

16 Aug 2019

PONE-D-19-18193

Plasma Trimethylamine-N-Oxide and Impaired Glucose Regulation: Results from The Oral Infections, Glucose Intolerance and Insulin Resistance Study (ORIGINS)

PLOS ONE

Dear Dr. Demmer,

Thank you for submitting your manuscript to PLOS ONE. After careful consideration, we feel that it has merit but does not fully meet PLOS ONE’s publication criteria as it currently stands. Therefore, we invite you to submit a revised version of the manuscript that addresses the points raised during the review process.

We would appreciate receiving your revised manuscript by Sep 30 2019 11:59PM. To enhance the reproducibility of your results, we recommend that if applicable you deposit your laboratory protocols in protocols.io, where a protocol can be assigned its own identifier (DOI) such that it can be cited independently in the future. For instructions see: http://journals.plos.org/plosone/s/submission-guidelines#loc-laboratory-protocols

We look forward to receiving your revised manuscript.

Kind regards,

Cheng Hu

Academic Editor

PLOS ONE

Journal Requirements:

1. Please provide additional details regarding participant consent. In the ethics statement in the Methods and online submission information, please ensure that you have specified what type of consent you obtained (for instance, written or verbal). If your study included minors, state whether you obtained consent from parents or guardians. If the need for consent was waived by the ethics committee, please include this information.

Reviewers' comments:

Reviewer's Responses to Questions

**Comments to the Author**

1. Is the manuscript technically sound, and do the data support the conclusions?

Reviewer #1: Partly

Reviewer #2: Partly

2. Has the statistical analysis been performed appropriately and rigorously? 

Reviewer #1: Yes

Reviewer #2: Yes

3. Have the authors made all data underlying the findings in their manuscript fully available?

Reviewer #1: No

Reviewer #2: Yes

4. Is the manuscript presented in an intelligible fashion and written in standard English?

Reviewer #1: Yes

Reviewer #2: Yes

5. Review Comments to the Author

Reviewer #1: Roy and colleagues present the relationship between plasma TMAO and early biomarkers of type 2 diabetes risk from ORIGINS based on the participants who are men and women aged 20-55 years, diabetes-free populations without a history of cardiovascular and/or kidney disease. The results are not hugely ground-breaking. However, overall, the manuscript is technically sound and the results are well interpreted.

There are some major and minor concerns that are summarized below and the authors need to address:

The authors need to provide the rationale to use the methods and statistical analysis in the body to appreciate why they were done (eg. why do the authors do a two-year follow-up visit to measure TMAO? Why do they use UPLC-MS/MS to measure TMAO?)

It would be helpful to the reader to make sure to explain what abbreviation means the first time they use it (eg. IQR, CARDIA and so on).

I would suggest that it is important to provide all data or refer to the data availability.

It would be helpful to improve the resolution of figures.

In Table 1, how does activity level estimate?

In Figure 2, the minimum would start 0.

Reviewer #2: The manuscript explores the possibility of using Trimethylamine-N-oxide (TMAO), as an early biomarker of longitudinal glucose increase or prevalent impaired glucose regulation. Overall, the manuscript is well-written, and the study has several interesting information. But few errors with lack of validation makes the study weak. These needs to be addressed sufficiently to be considered for publication in Plos Journal.

Comments:

1.Plasma TMAO in Diabetes, whether TMAO is a cause or consequence of Diabetes (Pre-diabetes) are highly debatable. As several studies using TMAO and cardiovascular diseases provide contradicting results. Though this showed modest increase in the prevalence of prediabetes in a nonlinear fashion. Several factors which interfere with the conditions/ Pre-diabetes were lest discussed and also, Study validation is an absolute necessity.

2. There are no line numbers in the manuscript which makes it hard to pinpoint the errors.

3. 20-55 years are definitely not young population, Correct the sentences wherever needed (Eg: Last two lines of introduction).

4. TMAO levels were modestly associated with increased age. Discuss more on this point.

6. PLOS authors have the option to publish the peer review history of their article (what does this mean?). If published, this will include your full peer review and any attached files.

Reviewer #1: No

Reviewer #2: No

---

## [Author Response · Author response to Decision Letter 0]

27 Sep 2019

Editor comments:

The body of the manuscript has been updated to comply with the formatting guidelines. A wrriten verbal consent was obtained from participants. This has been added to the Methods section of the manuscript as well as in the online submission section.

We wish to thank the reviewers for their helpful comments, which have led to further strengthening of the manuscript. We provide a point-by-point response to reviewer comments below.

Reviewer #1 Comments 

General comments: Roy and colleagues present the relationship between plasma TMAO and early biomarkers of type 2 diabetes risk from ORIGINS based on the participants who are men and women aged 20-55 years, diabetes-free populations without a history of cardiovascular and/or kidney disease. The results are not hugely ground-breaking. However, overall, the manuscript is technically sound and the results are well interpreted.

Response: We appreciate the reviewer’s comments and have responded to each specific comment below in a point-by-point fashion.

Comment #1: The authors need to provide the rationale to use the methods and statistical analysis in the body to appreciate why they were done (eg. why do the authors do a two-year follow-up visit to measure TMAO? Why do they use UPLC-MS/MS to measure TMAO?)

Response: We wish to clarify that TMAO was only measured once at baseline in this study. This is stated on page 6 (line 117) of our revised submission. 

In this analysis, we were interested in investigating the following questions:

a. Is baseline plasma TMAO levels associated with insulin resistance, fasting glucose levels and HbA1c cross-sectionally

b. Is baseline plasma TMAO levels associated with longitudinal increase in fasting plasma glucose

c. Is baseline plasma TMAO levels are associated with increased prediabetes prevalence cross-sectionally

To our knowledge there are no data addressing the association between baseline TMAO and longitudinal fasting plasma glucose among generally healthy, diabetes-free population.

Traditional assays for choline metabolites such as TMAO, including radioenzymatic assays, liquid chromatography with electrochemical detection and GCMS are characterized by low specificity and tedious sample preparation with multi-step derivatization protocols. Targeted metabolomic approaches employing Liquid Chromatography-tandem mass spectrometry (UPLC-MS/MS) platforms under multiple reaction mode offer high sensitivity, specificity, accuracy, and precision for the quantitation of small molecules. The UPLC-MS/MS based assay used in the present study enabled the sensitive and accurate measurement of TMAO in the human samples. The assay has a lower limit of quantitation (LLOQ) of 0.05uM. The intra-assay accuracy and precision 98% and 2.78% while inter-assay accuracy and precision was 97% and 4.11%.

Comment #2: It would be helpful to the reader to make sure to explain what abbreviation means the first time they use it (eg. IQR, CARDIA and so on).

Response: We have now clarified abbreviations of IQR on page 8 (line 180) and CARDIA on page 14 (line 267) are spelled out as it first appears in the paper.

Comment #3: I would suggest that it is important to provide all data or refer to the data availability.

Response: All relevant data are within the manuscript and a limited data set is privately available on Figshare and a doi has been issued (10.6084/m9.figshare.9913667)

Comment #4: It would be helpful to improve the resolution of figures.

Response: The resolution of figures is now improved and revised figures are uploaded.

Comment #5: In Table 1, how does activity level estimate?

Response: The methods used to quantify Leisure-time physical activity are described on page 7 (line number 145) of the revision. 

Comment #6: In Figure 2, the minimum would start 0.

Response: Since a value of zero is arbitrary relative to the mean values of fasting glucose and the variability of the measure, we have chosen to center the figure Y axis on the mean value observed in the total population (mean=85 mg/dL) and to set the range as twice the standard deviation of fasting glucose (14) in this sample. This gives the reader a better perspective of glucose variability across TMAO groups relative to the size of glucose variability overall. This is now clarified in the figure legend. 

Reviewer # 2 Comments

General comments: The manuscript explores the possibility of using Trimethylamine-N-oxide (TMAO), as an early biomarker of longitudinal glucose increase or prevalent impaired glucose regulation. Overall, the manuscript is well-written, and the study has several interesting information. But few errors with lack of validation makes the study weak. These needs to be addressed sufficiently to be considered for publication in Plos Journal.

Response: We appreciate the reviewer’s comments and have addressed each of the comments below.

Comments # 1: Plasma TMAO in Diabetes, whether TMAO is a cause or consequence of Diabetes (Pre-diabetes) are highly debatable. As several studies using TMAO and cardiovascular diseases provide contradicting results. Though this showed modest increase in the prevalence of prediabetes in a nonlinear fashion. Several factors which interfere with the conditions/ Pre-diabetes were lest discussed and also, Study validation is an absolute necessity.

Response: We agree that results from previous studies have mixed results in the association of TMAO and cardiovascular disease. We were able to replicate the null findings that were observed between TMAO and cross-sectional insulin resistance in the CARDIA study by Meyer et al*, the latter being a generally healthy population similar to ORIGINS. While our data cannot prove causality, they help to characterize TMAO in relation to early cardiometabolic risk which will help to inform the design and premise of future studies that can address causal associations with greater validity.

*Meyer KA, Benton TZ, Bennett BJ, Jacobs DR, Jr., Lloyd-Jones DM, Gross MD, et al. Microbiota-Dependent Metabolite Trimethylamine N-Oxide and Coronary Artery Calcium in the Coronary Artery Risk Development in Young Adults Study (CARDIA). Journal of the American Heart Association. 2016;5(10).

Comment # 2: There are no line numbers in the manuscript which makes it hard to pinpoint the errors.

Response: Line numbers are now added to facilitate the review

Comment # 3: 20-55 years are definitely not young population, Correct the sentences wherever needed (Eg: Last two lines of introduction).

Response: Reference to 20-55 years as young population was removed from the following lines (line 92 on page 5, line 318 and line 321 on page 16).

Comment #4: TMAO levels were modestly associated with increased age. Discuss more on this point.

Response: Table 1 shows the mean age of those in Tertile 3 is 36.01±1.03 compared to Tertile 1 which was 32.28±0.92. The reasons for this relationship are not entirely clear but one possibility is that older participants have decreased renal function leading to increased TMAO. In response to the reviewer’s comments, we have added more discussion on this point to the manuscript on page 14 as follows: “Renal function might also provide some level of explanation for discordant findings in our current data as compared to other cohorts. The fact that higher renal function increases TMAO clearance raises the strong potential for confounding. Specifically, it is possible that reduced renal function causes both elevated TMAO levels and clinical CVD events. Our observation of increased TMAO among older participants provides indirect support for this notion. As such, our adjustment for age in multivariable models helps to mitigate confounding by renal function although future studies with assessment of renal function will be important.”

---

## [Decision Letter · Decision Letter 1]

26 Nov 2019

PONE-D-19-18193R1

Plasma Trimethylamine-N-Oxide and Impaired Glucose Regulation: Results from The Oral Infections, Glucose Intolerance and Insulin Resistance Study (ORIGINS)

PLOS ONE

Dear Dr. Demmer,

Thank you for submitting your manuscript to PLOS ONE. After careful consideration, we feel that it has merit but does not fully meet PLOS ONE’s publication criteria as it currently stands. Therefore, we invite you to submit a revised version of the manuscript that addresses the points raised during the review process.

We would appreciate receiving your revised manuscript by Jan 10 2020 11:59PM. To enhance the reproducibility of your results, we recommend that if applicable you deposit your laboratory protocols in protocols.io, where a protocol can be assigned its own identifier (DOI) such that it can be cited independently in the future. For instructions see: http://journals.plos.org/plosone/s/submission-guidelines#loc-laboratory-protocols

We look forward to receiving your revised manuscript.

Kind regards,

Cheng Hu

Academic Editor

PLOS ONE

Reviewers' comments:

Reviewer's Responses to Questions

**Comments to the Author**

1. If the authors have adequately addressed your comments raised in a previous round of review and you feel that this manuscript is now acceptable for publication, you may indicate that here to bypass the “Comments to the Author” section, enter your conflict of interest statement in the “Confidential to Editor” section, and submit your "Accept" recommendation.

Reviewer #1: All comments have been addressed

2. Is the manuscript technically sound, and do the data support the conclusions?

Reviewer #1: Yes

3. Has the statistical analysis been performed appropriately and rigorously? 

Reviewer #1: Yes

4. Have the authors made all data underlying the findings in their manuscript fully available?

Reviewer #1: Yes

5. Is the manuscript presented in an intelligible fashion and written in standard English?

Reviewer #1: Yes

6. Review Comments to the Author

Reviewer #1: I would suggest that the manuscript needs to provide all abbreviations used.

Page 3 line 6 CRP -> C-reactive protein

Page 5 line 105 FPG -> fasting plasma glucose (FPG)

Page 5 line 106 HbA1c -> hemoglobin A1c (HbA1c)

Page 6 line 124 fasting plasma glucose (FPG) -> FPG

Page 8 line 169 (PR) -> Delete

Page 8 line 182 Trimethylamine-N-oxide (TMAO) -> TMAO

Page 8 line 183 The Oral Infections, Glucose intolerance and Insulin Resistance Study (ORIGNS) -> ORIGINS

Page 11 line 211 hemoglobin A1c -> HbA1c

Page 11 line 212 Trimethylamine-N-oxide (TMAO) -> TMAO

Page 11 line 213 The Oral Infections, Glucose intolerance and Insulin Resistance Study (ORIGNS) -> ORIGINS

Page 14 line 276 CVD -> cardiovascular disease

Page 15 line 284 CKD -> chronic kidney disease

Page 15 line 301 Alternative Healthy Eating Index (AHEI) -> AHEI

It would be helpful to the reader to provide the rationale in the figure 2 legend which the authors explained in the response of comment 6.

7. PLOS authors have the option to publish the peer review history of their article (what does this mean?). If published, this will include your full peer review and any attached files.

Reviewer #1: No

---

## [Author Response · Author response to Decision Letter 1]

27 Nov 2019

We wish to thank the reviewer for the helpful comments, which have led to further strengthening of the manuscript. 

Reviewer #1 Comments 

Reviewer #1: I would suggest that the manuscript needs to provide all abbreviations used.

Page 3 line 6 CRP -> C-reactive protein

Page 5 line 105 FPG -> fasting plasma glucose (FPG)

Page 5 line 106 HbA1c -> hemoglobin A1c (HbA1c)

Page 6 line 124 fasting plasma glucose (FPG) -> FPG

Page 8 line 169 (PR) -> Delete

Page 8 line 182 Trimethylamine-N-oxide (TMAO) -> TMAO

Page 8 line 183 The Oral Infections, Glucose intolerance and Insulin Resistance Study (ORIGNS) -> ORIGINS

Page 11 line 211 hemoglobin A1c -> HbA1c

Page 11 line 212 Trimethylamine-N-oxide (TMAO) -> TMAO

Page 11 line 213 The Oral Infections, Glucose intolerance and Insulin Resistance Study (ORIGNS) -> ORIGINS

Page 14 line 276 CVD -> cardiovascular disease

Page 15 line 284 CKD -> chronic kidney disease

Page 15 line 301 Alternative Healthy Eating Index (AHEI) -> AHEI

It would be helpful to the reader to provide the rationale in the figure 2 legend which the authors explained in the response of comment 6.

Response: We appreciate the reviewer’s comments and have provided the expansion/acronyms as appropriate in the corresponding pages as noted below:

Page 3 line 68 CRP now reads as C-reactive protein (CRP)

Page 5 line 105 FPG now reads as fasting plasma glucose (FPG)

Page 5 line 106 HbA1c now reads as hemoglobin A1c (HbA1c)

Page 6 line 124 fasting plasma glucose is deleted and now reads as FPG

Page 8 line 169 (PR) is deleted

Page 8 line 182 Trimethylamine-N-oxide (TMAO) now reads as TMAO

Page 8 line 183 The Oral Infections, Glucose intolerance and Insulin Resistance Study (ORIGNS) now reads as ORIGINS

Page 11 line 211 hemoglobin A1c now reads as HbA1c

Page 11 line 212 Trimethylamine-N-oxide (TMAO) now reads as TMAO

Page 11 line 213 The Oral Infections, Glucose intolerance and Insulin Resistance Study (ORIGNS) now reads as ORIGINS

Page 14 line 276 CVD now reads as cardiovascular disease

Page 15 line 284 CKD now reads as chronic kidney disease

Page 15 line 301 Alternative Healthy Eating Index (AHEI) now reads as AHEI

The rationale that was explained in the comment 6 has been added to the figure 2. legend in the revised manuscript text.

---

## [Decision Letter · Decision Letter 2]

20 Dec 2019

Plasma Trimethylamine-N-Oxide and Impaired Glucose Regulation: Results from The Oral Infections, Glucose Intolerance and Insulin Resistance Study (ORIGINS)

PONE-D-19-18193R2

Dear Dr. Demmer,

We are pleased to inform you that your manuscript has been judged scientifically suitable for publication and will be formally accepted for publication once it complies with all outstanding technical requirements.

With kind regards,

Cheng Hu

Academic Editor

PLOS ONE

Additional Editor Comments (optional):

Reviewers' comments:

Reviewer's Responses to Questions

**Comments to the Author**

1. If the authors have adequately addressed your comments raised in a previous round of review and you feel that this manuscript is now acceptable for publication, you may indicate that here to bypass the “Comments to the Author” section, enter your conflict of interest statement in the “Confidential to Editor” section, and submit your "Accept" recommendation.

Reviewer #1: All comments have been addressed

2. Is the manuscript technically sound, and do the data support the conclusions?

Reviewer #1: Yes

3. Has the statistical analysis been performed appropriately and rigorously? 

Reviewer #1: Yes

4. Have the authors made all data underlying the findings in their manuscript fully available?

Reviewer #1: Yes

5. Is the manuscript presented in an intelligible fashion and written in standard English?

Reviewer #1: Yes

6. Review Comments to the Author

Reviewer #1: (No Response)

7. PLOS authors have the option to publish the peer review history of their article (what does this mean?). If published, this will include your full peer review and any attached files.

Reviewer #1: No

---

## [Editor Report · Acceptance letter]

3 Jan 2020

PONE-D-19-18193R2 

Plasma Trimethylamine-N-Oxide and Impaired Glucose Regulation: Results from The Oral Infections, Glucose Intolerance and Insulin Resistance Study (ORIGINS) 

Dear Dr. Demmer:

I am pleased to inform you that your manuscript has been deemed suitable for publication in PLOS ONE. Congratulations! Your manuscript is now with our production department. 

With kind regards,

on behalf of

Dr. Cheng Hu 

Academic Editor

PLOS ONE